# Study of Neuroprotection by a Combination of the Biological Antioxidant (*Eucalyptus* Extract) and the Antihypertensive Drug Candesartan against Chronic Cerebral Ischemia in Rats

**DOI:** 10.3390/molecules26040839

**Published:** 2021-02-05

**Authors:** Christine Trabolsi, Wafaa Takash Chamoun, Akram Hijazi, Cendrine Nicoletti, Marc Maresca, Mohamad Nasser

**Affiliations:** 1Neuroscience Research Center, Faculty of Medical Sciences, Lebanese University, Beirut P.O. Box 6573/14, Lebanon; christinetrabolsi9@gmail.com (C.T.); wafaa.takash@ul.edu.lb (W.T.C.); 2Rammal Hassan Rammal Research Laboratory, Physiotoxicity (PhyTox), Faculty of Sciences, Lebanese University, Beirut P.O. Box 6573/14, Lebanon; 3Plateforme de recherche et d’analyse en sciences de l’environnement (EDST-PRASE), Beirut P.O. Box 6573/14, Lebanon; Akram.Hijazi@ul.edu.lb; 4Aix Marseille University, CNRS, Centrale Marseille, iSm2, 13397 Marseille, France; cendrine.nicoletti@univ-amu.fr

**Keywords:** chronic cerebral ischemia, Candesartan, *Eucalyptus*, rats

## Abstract

Chronic cerebral ischemia with a notable long-term cessation of blood supply to the brain tissues leads to sensorimotor defects and short- and long-term memory problems. Neuroprotective agents are used in an attempt to save ischemic neurons from necrosis and apoptosis, such as the antioxidant agent *Eucalyptus*. Numerous studies have demonstrated the involvement of the renin-angiotensin system in the initiation and progression of cardiovascular and neurodegenerative diseases. Candesartan is a drug that acts as an angiotensin II receptor 1 blocker. We established a rat model exhibiting sensorimotor and cognitive impairments due to chronic cerebral ischemia induced by the ligation of the right common carotid artery. Wistar male rats were randomly divided into five groups: Sham group, Untreated Ligated group, Ischemic group treated with *Eucalyptus* (500 mg/kg), Ischemic group treated with Candesartan (0.5 mg/kg), and Ischemic group treated with a combination of *Eucalyptus* and Candesartan. To evaluate the sensorimotor disorders, we performed the beam balance test, the beam walking test, and the modified sticky test. Moreover, the object recognition test and the Morris water maze test were performed to assess the memory disorders of the rats. The infarct rat brain regions were subsequently stained using the triphenyltetrazolium chloride staining technique. The rats in the Sham group had normal sensorimotor and cognitive functions without the appearance of microscopic ischemic brain lesions. In parallel, the untreated Ischemic group showed severe impaired neurological functions with the presence of considerable brain infarctions. The treatment of the Ischemic group with a combination of both *Eucalyptus* and Candesartan was more efficient in improving the sensorimotor and cognitive deficits (*p* < 0.001) than the treatment with *Eucalyptus* or Candesartan alone (*p* < 0.05), by the comparison to the non-treated Ischemic group. Our study shows that the combination of *Eucalyptus* and Candesartan could decrease ischemic brain injury and improve neurological outcomes.

## 1. Introduction

Stroke is defined as a sudden cessation of blood supply to brain tissue resulting from hemorrhagic or ischemic disease, causing severe neurological impairment [1]. Ischemic strokes occur when the arteries of the brain shrink or become blocked, resulting in a significant decrease in blood flow (ischemia). In fact, cerebral ischemia is one of the leading causes of death in the world, it is also the cause of sensorimotor defects and short- and long-term memory problems [2], as well as many neurodegenerative disorders [3]. Thus, the neuroprotection against the cerebral ischemia is highly needed [4]. On the other hand, the renin angiotensin aldosterone system (RAAS) is one of the major regulatory systems that controls blood pressure, cardiovascular function, and fluid balance in the body [5]. RAAS is involved in the initiation and progression of many pathologies, such as arterial hypertension, cardiovascular diseases, and several neurodegenerative disorders, such as the stroke [6]. The components of RAAS are also present in the brain structures involved in cognition, behavior, and locomotion [7]. In addition, there is considerable evidence showing the potential function of angiotensin II in the etiology of certain neurodegenerative diseases, and the development of metabolic syndrome [7]. The deleterious effects of the RAAS system are mainly mediated by the angiotensin 1 receptor (AT1R) pathway of angiotensin II [8]. Candesartan is indeed an AT1R antagonist that has been used for neuroprotection in animal models of stroke [9]. Moreover, several studies revealed the antioxidant neuroprotective role of *Eucalyptus* [10]. In fact, some in vitro and in vivo studies show that *Eucalyptus* has anti-inflammatory, anticancer [11], antihypertensive, antibacterial [12], antiviral, and antifungal properties [13]. *Eucalyptus* leaf extracts were effective in ameliorating hydrogen peroxide-induced oxidative stress by increasing cell viability, glutathione levels, and antioxidant enzymes activity, and by decreasing the free radicals production and lipid peroxidation levels [14]. *Eucalyptus* is used in Europe as a traditional remedy for inflammation-related disorders such as arthritis, diabetes, and asthma [15]. The major goals of our present research work were first to study the neuroprotective effect(s) of the combination of *Eucalyptus Camaldulensis* leaf extracts and Candesartan (AT1R) against cerebral ischemia induced by chronic unilateral ligation of the right common carotid artery (RCCA) in the rat, and second to investigate for a potential neuroprotective role of the *Eucalyptus* plant.

## 2. Results

### 2.1. Body Weight

The weight of the animal was measured throughout the days of the experiment, and the variation of its mass was measured according to the formula:ΔM = (Mf − Mi)/Mi, (Mf = mass at day +16, Mi = mass at day − 1).

A weight loss was observed in the Ischemic group (average ΔM = −0.1924), Combination group (average ΔM = −0.1308), Candesartan group (average ΔM = −0.1581), and *Eucalyptus* group (average ΔM = −0.1261) by the end of the experiment, with a significant difference (*p* < 0.05) compared to the Sham group (average ΔM = 0.138) that showed a gain of weight (positive ΔM).

### 2.2. Chemical Testing Results

#### 2.2.1. Yields of *Eucalyptus Camaldulensis* Crude Extracts

Table 1 shows the yield percentages of active compounds extracted from *Eucalyptus Camaldulensis*. The aqueous solvent has more ability than the ethanolic extract to extract the active components from the plant, so we can use it in our study.

#### 2.2.2. Phytochemical Screening Tests

Table 2 demonstrates the phytochemical screenings of the two extracts (ethanolic and aqueous) of *Eucalyptus Camaldulensis*. As shown in the table below, there are qualitative differences between the ethanolic and aqueous extracts in terms of presence (+) or absence (−) of resins, cardiac glycosides, and as well as saponins. Important medicinal phytochemicals such as phenols, terpenoids, flavonoids, alkaloids, sterols and steroids, fixed oil, and fatty acids are present in *Eucalyptus Camaldulensis*.

### 2.3. Chemical Quantifications of Secondary Products

The total phenolic content (TPC) of both ethanolic and aqueous extracts of *Eucalyptus Camaldulensis* were estimated, where the aqueous extract contains more phenols than the ethanolic one (Table 3).

### 2.4. Antioxidant Activity (AA)

The transformation of the violet color to a yellow color indicates a high AA, where the aqueous extract has a higher AA compared to the ethanolic one (Table 3).

### 2.5. Behavioral and Cognitive Test Results

#### 2.5.1. Beam Balance Test (BBT)

Prior to the surgical operation at D (day)–1, all animal groups exhibited a score of 1, showing a stable balance on the beam. This score increased in all experimental groups to 1.49, 2.6, 1.69, 2.53, and 2.43 respectively, reflecting an unstable balance with a significant change between the Sham and Ischemic groups (*p* < 0.05) (Figure 1). This score started to decrease in the Sham and all treated groups to reach the normal one at D + 7, while the Ischemic group continued to worsen and reached a score of 3.3 where the rats tried to balance by hanging through 1 or 2 limbs. Regarding the neurological score, a significant change was shown between the Sham and the following groups: Ischemic, Candesartan, and *Eucalyptus, p* < 0.001, *p* < 0.05, and *p* < 0.01 respectively, as well as between Ischemic and Combination groups (*p* < 0.001), Candesartan group (*p* < 0.001), and *Eucalyptus* group (*p* < 0.01). By comparing the 5 groups at different days, at D + 3, we observed a significant difference between Sham and Ischemic groups (*p* < 0.001), Candesartan groups (*p* < 0.05), *Eucalyptus* groups (*p* < 0.01), as well as between the Ischemic and Combination groups (*p* < 0.01). The combination treatment continued to be effective at D + 5, as shown by a significant difference between the Ischemic and the Combination groups (*p* < 0.001), also the treatment with Candesartan alone and *Eucalyptus* alone appeared to be efficient, with a significant difference between ischemic and Candesartan groups (*p* < 0.001) and between Ischemic and *Eucalyptus* groups (*p* < 0.01). There was also a significant change between Sham and Ischemic groups (*p* < 0.001), Candesartan group (*p* < 0.01), and *Eucalyptus* group (*p* < 0.001).

All treatments showed their effectiveness on D + 7, with a significant difference between Ischemic and all treated groups (*p* < 0.001).

#### 2.5.2. Beam Walking Test (BWT) Score

Prior to surgical operation, concerning the neurological score results, at D − 1, no significant difference was noticed between all experimental groups on the beam walking apparatus, and the same was noted at D + 3, D + 5, and D + 7. Only at D + 1 did the difference appear to be significant between Ischemic and Sham groups (*p* < 0.05). Concerning the scores taken while rats traverse the beam, they appeared to be strongly increased at D + 1, but they started to decrease gradually to reach nearly the normal measurements, such as those at days prior to surgery (Figure 2).

#### 2.5.3. Beam Walking Test Time

Before the operation, all rats crossed the beam quickly within 4 to 6 s. The surgeries slowed the rats speed so that they needed more time to traverse the beam in all experimental groups, except the Sham at D + 1, D + 3, and D + 5, with a significant difference between the Ischemic and Sham groups (*p* < 0.001, *p* < 0.01, and *p* < 0.05) at D + 1, D + 3, and D + 5, respectively (Figure 3). From D + 5, the average time needed to traverse the beam decreased to reach nearly the normal value at D + 7 in all experimental groups with no significant change. Also, at D + 1, we noticed a significant difference between Sham and Combination groups (*p* < 0.001), between the Ischemic and Candesartan groups (*p* < 0.05), between the Ischemic and *Eucalyptus* groups (*p* < 0.05), between Combination and Candesartan groups (*p* < 0.01), and between Combination and *Eucalyptus* groups (*p* < 0.01).

#### 2.5.4. Modified Sticky Tape Test (MSTT)

MSTT was applied to all groups (Sham, Ischemic, Ischemic treated with a combination of Candesartan and *Eucalyptus*, Ischemic treated with Candesartan, and Ischemic treated with *Eucalyptus*) (Figure 4). At D − 1, nearly all rats showed a ratio 1 of left to right limb time sensation. At D + 1, all groups showed a decrease in the ratio from 1 to 0.99, 0.64, 0.88, 0.74, and 0.58, respectively. This ratio increased in all groups at D + 3 to 0.77 (Ischemic), 0.91 (Combination), 0.93 (Candesartan), and 0.85 (*Eucalyptus*), while it remained constant in the Sham group. As the time passed from D + 3 to +7, the ratio showed a slight increase in all groups to 1.02, 0.83, 0.96, 0.93, and 0.95, respectively. There was a significant difference between Sham and Ischemic groups (*p* < 0.001), and between Sham and *Eucalyptus* groups (*p* < 0.05), and also between the Ischemic and Combination groups (*p* < 0.01). By comparing the 5 groups at different days, a significant difference was detected at D + 1 between Ischemic and Sham groups (*p* < 0.001), Ischemic and Combination groups (*p* < 0.01), Sham and *Eucalyptus* groups (*p* < 0.01), and Combination and *Eucalyptus* groups (*p* < 0.05). Similarly, a significant difference was noted at D + 3 between Sham and Ischemic groups (*p* < 0.05). However, as we noticed in the graph above, no significant changes were presented at D − 1, D + 5, and D + 7.

#### 2.5.5. Object Recognition Test (ORT)

Novel/familiar (N/F) objects ratio represents the time taken by the rat exploring a novel object to that of the previously discovered familiar object. The highest ratio was recorded for the Combination group (Ratio = 6.67), and the lowest for Ischemic group (Ratio = 1.02). There was a significant change between the Combination and Ischemic groups (*p* < 0.05), between Combination and Candesartan groups (*p* < 0.05), and between Combination and *Eucalyptus* groups (*p* < 0.05) as well (Figure 5).

#### 2.5.6. Morris Water Maze Test (MWMT)

MWMT was first done at D + 10, when nearly all groups took time to detect the place of the hidden platform. Going to the end of the test after 7 days, all treated groups showed an improvement in their performance, with a latency of respectively 9.67, 18.89, and 13.41 for the Ischemic group treated with a combination of Candesartan and *Eucalyptus*, Candesartan alone, and *Eucalyptus* alone (Figure 6). In parallel, the Ischemic group had a latency of 42.99 and was the slowest to improve. A significant difference between the Ischemic group and all treated groups (*p* < 0.001) and between the Ischemic and Sham groups (*p* < 0.001) showed that the ligation of the RCCA impaired the overall performance of the rat, including learning and spatial memory in contrast to the treatment that improved it. By comparing the five experimental groups at different days, we observed a significant difference between the Ischemic and Sham groups from D + 11 until D + 16 (*p* < 0.05), at D + 11, at D + 12, at D + 13 (*p* < 0.01), and at D + 14, at D + 15, and at D + 16 (*p* < 0.01). Interestingly, the treatment with a combination of Candesartan and *Eucalyptus* started to be effective at D + 11 (*p* < 0.05) and continued at D + 13 (*p* < 0.05), at D + 14 (*p* < 0.001), at D + 15, and at D + 16 (*p* < 0.01). The treatments with *Eucalyptus* alone and Candesartan alone showed their efficacy starting from D + 14 (*p* < 0.01) until D + 16 (*p* < 0.05). The treatments by Candesartan alone or *Eucalyptus* alone improved the rat performance, but it became less efficient than the combination of Candesartan and *Eucalyptus*.

#### 2.5.7. Triphenyltetrazolium Chloride (TTC) Test

TTC is an effective and easy method to detect by staining the infarcted rat brain tissue regions. TTC stains normal areas of the brain red. Brain sections of all groups were stained red except for sections of the Ischemic brains, where white colored areas were observed in all animals at the infarcted areas or sites of damaged tissues (area of 4.66 ± 0.615 mm^2^) (Figure 7 and Figure 8). TTC staining clearly demonstrated that both Candesartan alone, *Eucalyptus* alone, and the combination of Candesartan and *Eucalyptus* protected the brains of rats from tissue damage caused by ischemia. Interestingly, no white areas were observed in the brains of those groups, in contrast to the Ischemic group.

## 3. Discussion

Stroke continues to be an extremely prevalent disease and poses a great challenge in the development of safe and effective therapeutic options [1]. A number of neurodegenerative diseases are associated with hypoxic and ischemic cases characterized by decreased oxygen supply to brain tissue [16].

In our present study, we established a rat model exhibiting a sensorimotor deficit and cognitive impairment under chronic cerebral ischemia induced by the RCCA ligation. Our pilot study shed light on the neuroprotection against the treatment of cerebral ischemia by combining an antihypertensive drug Candesartan and the medicinal plant *Eucalyptus Camaldulensis*.

According to our results of all performed tests (BBT, BWT, and MSTT), RCCA ligation led to a sensorimotor deficit in the left side of the animal body, which is called a counter-lateral deficit. In fact, BBT results of all treated groups showed that the treatment with a combination of both Candesartan and *Eucalyptus* was more efficient to ameliorate the neurological score than with Candesartan alone and *Eucalyptus* alone, with a difference starting from D + 3 to D + 7. While MSTT showed, with the experimental groups of BBT, similar results, but with a difference starting and ending at day +1 throughout the seven post-surgical days, where the treated rats might have felt the presence of tape on their carp more compared to untreated rats. In addition, in the sensorimotor test BWT, the treatments of Candesartan alone and *Eucalyptus* alone were efficient to improve the deterioration at the level of recorded time only at D + 1 and D + 3. In the three sensorimotor tests, all groups started to relieve slowly on the last days of the tests to reach nearly the normal value, as before surgery, except for the Ischemic group, where the neurological score continued to increase in BBT. This observation indicated that the ligation of the RCCA led to a malfunction in the cerebral hemisphere, more precisely at the area that is responsible for sensation and reflex of the left front limb, while the treatments and especially the combined one restored this function.

Motor impairment of stroke patients could be explained as a dysfunction of ipsilesional motor areas leading to deficits in motor control, or another explanation that might be acceptable is that transcallosal inhibition occurs (e.g., the ipsilesional hemisphere is exposed to pathologically enhanced inhibitory influences exerted by contralesional hemisphere), which leads to the decreased motor performance [17]. At the acute phase of stroke, different changes at the molecular and network levels accelerated the brain self-repair processes [18]. Previous studies had shown that earlier rehabilitation is more critical at the level of motor functional abilities [19]. This made it important for further studies to investigate the neural mechanisms involved in early-stage motor deficits in order to provide important theoretical evidence for early neurorehabilitation of ischemic stroke [20].

Also, in stroke animals, such improvements in behavioral performance patterns could be facilitated by behavioral training, but the time course of post-stroke recovery in animals is typically shorter [21].

However, the spontaneous motor recovery in stroke patients was mostly realized in the first six weeks after stroke [22]. Our data demonstrated that at the level of the Ischemic group, and after the deterioration of motor performance post-stroke, a slight recovery was shown in the BWT and MSTT results, with no significant difference with the Sham group. While in BBT, the effect of the ligation continued to be severe in the Ischemic group. This suggests that the recovery might be accomplished during a larger time interval post-ligation. In addition, our data at the level of cognitive tests showed a significant cognitive deficit at the level of the Ischemic group in MWMT results, but that was not seen in ORT results. However, considering MWMT results, we observed a gradual improvement of cognitive performance with time in the Ischemic group, and that could be explained in terms of functional recovery, but the learning in this group was slow. Moreover, we found that the treatment with a combination of Candesartan and *Eucalyptus* enhanced the spatial learning and memory of rats starting from D + 11, while the treatment of Candesartan or *Eucalyptus* alone started later, being efficient from D + 14.

Reduced cognitive function is closely related to ischemic stroke. Previous studies correlated cerebral ischemia-induced hippocampal damage and spatial learning deficits and found that the hippocampal CA1 region is involved in cognitive function for information processing, and that the hippocampal CA3 region contains glutamate receptors (N-methyl-D-aspartate NMDA receptors) which play a role in associative memory [23]. These hippocampal regions are also shown to play an important role in encoding new spatial information within short-term memory. A recent study showed the effect of laser acupuncture to attenuate memory impairment and facilitate recovery by improving neural density in the hippocampus, showing that it had been reduced in number after cerebral ischemia. This is achieved by laser acupuncture stimulation that increases cAMP response element-binding (CREB) activity by reducing the neuronal cell loss at the hippocampus [24]. Knowing that, after ischemic stroke, it was shown that neurons necrotize in the infarct core; however, in the surrounding region (the penumbra area), they took hours to days to die after ischemia onset, which provides a possible therapeutic target to rescue the injured neurons from autophagy and necrosis. Recent studies have shown that recovery after stroke can be modulated by neuromodulation techniques, such as repetitive transcranial magnetic stimulation and transcranial direct current stimulation [22], and that angiogenesis plays a critical role in such recovery [25].

Chronic cerebral ischemia involves many pathological characteristics that include mainly oxidative stress, inflammation, and apoptosis [26]. Therefore, agents with antioxidant, anti-apoptotic, and anti-inflammatory properties would be recommended to treat it [27]. In the present study, we used a combination therapy of *Eucalyptus* leaves extracts and Candesartan to treat our rat model after chronic RCCA ligation. It was obvious throughout all the tests that the combination therapy improved the case and reduced the stroke severity.

According to previous studies, Candesartan treatment has been shown to improve sensorimotor and cognitive function after traumatic brain injury (TBI) in mice and tested by rotarod and MWMT [28]. Studies show that Aliskiren, Enalapril, and Candesartan significantly improve spatial learning and memory and inhibit hippocampal apoptosis [29]. In a model with traumatic brain injury in mice, the results indicated that Candesartan is neuroprotective by reducing neuronal damage, decreasing lesion volume and microglial activation, protecting cerebrospinal fluid, and improving functional behavior [28]. Excessive production of superoxide after cerebral ischemia is known to cause neuronal injury.

The activation of AT1R leads to the production of superoxide [30], but its blockade by Candesartan prevents superoxide production.

Approximately 30% of CA1 neurons in the hippocampus survived in Candesartan-treated animals [31]. This neuroprotective effect of Candesartan was reported in our tests, but the combination of Candesartan and *Eucalyptus* has been shown to be more effective and beneficial in specific tests.

Also, studies demonstrated the beneficial effects of *Eucalyptus* extracts being able to detoxify the liberated free radicals, acting as an antioxidant, which made it become approved as a therapeutic target [32].

Many recent studies have been conducted on plants used in traditional medicine for treating various types of cancers, and antitumor compounds have been found in them. As an illustration, in vitro studies proved that the ethanolic extract of *Eucalyptus Camaldulensis* leaf exerts a cytotoxic effect on human lung and breast cancer [33].

Concerning the cytotoxic effect of essential oils of *Eucalyptus*, data is remarkably restricted.

In our study, qualitative and quantitative chemical analyses of active constituents of both aqueous and ethanolic extracts were performed. Then, the antioxidant activities of the ethanolic and aqueous extracts of *Eucalyptus Camaldulensis* were evaluated.

The phytochemical screening plays a good role in the estimation of possible involvement of this plant in the prevention or treatment of some diseases. In our present study, qualitative analysis showed that there is a qualitative difference between the ethanolic and aqueous extracts in terms of resins, cardiac glycosides, and saponins. Important medicinal phytochemicals such as phenols, terpenoids, flavonoids, alkaloids, sterols and steroids, fixed oils, and fatty acids are present in *Eucalyptus Camaldulensis*. Terpenoids are anti-inflammatory, antiviral, antibacterial, and inhibit cholesterol synthesis [34]. Flavonoids are antiallergenic, antiviral, anti-inflammatory, vasodilating, antioxidant, antimicrobial, photoreceptor, and have repellent and light-screening actions [35]. Phenols have antioxidant, antimicrobial, and anticancer activities [36]. These compounds have a neuroprotective activity against oxidative stress. Studies showed that the administration of Resveratrol, a polyphenolic compound, protected the damage caused by spinal cord ischemia in the rat [37].

Moreover, our results also showed that the aqueous extract of the *Eucalyptus* had more content of phenols and flavonoids and higher 1,1-diphenyl-2-picrylhydrazyl (DPPH) radical scavenging activity than the ethanolic extract. Therefore, we used the aqueous extract in our experiment as a treatment for the cerebral ischemia. We suggest that its combination with Candesartan could facilitate the work of the drug through its antioxidant function.

On the other hand, Curcumin is a natural polyphenol used in ancient Asian medicine [38]. The combined effects of Curcumin (a natural phenol) and Candesartan have been studied in cerebral ischemia induced by middle cerebral artery occlusion. The results of this study indicated that Curcumin synergistically enhances the inhibitory action of Candesartan on cerebral ischemia by suppressing changes in blood flow and oxidative stress via antioxidant properties, suggesting beneficial combined effects of Curcumin and Candesartan on ischemic brain injury [39].

The brain damage following RCCA ligation could be evaluated in different ways. The extent of the infarct is usually performed by TTC staining. TTC stains normal areas of the brain red. Our results showed that the brain sections of all groups were stained red, except for sections of the ischemic brain, where it was observed that the infarcted area is white in color, indicating the site of the damaged tissue. In one study, TTC was used to detect experimental cerebral ischemia, but its accuracy has not been fully established, particularly in early cerebral ischemia models [40]. In a model of carotid artery ligation in the mouse, TTC measurement results indicated that the histological damage reached the maximum 24 h after stroke [41]. TTC staining could also be performed accurately up to seven days after the stroke [42]. TTC staining showed that, as expected, Candesartan alone protects the rat’s brain from ischemic damage. Similarly, *Eucalyptus* alone and the combination of Candesartan and *Eucalyptus* are also able to protect the brain of rats from tissue injuries caused by ischemia.

## 4. Materials and Methods

### 4.1. Study Location

All the experimental procedures of the present work were performed at Rammal Hassan Rammal Research Laboratory, in the Lebanese University campus in Nabatieh, South Lebanon.

The present study was carried out after obtaining ethical clearance from the institutional ethics committee. The approval number is UL/2019/25661.

### 4.2. Experimental Models

A total of 37 Wistar male rats with a mean weight of 180–280 g were used during our study. Rats were randomly divided into 5 groups with varied numbers: Sham group (number (n) = 6 rats), Ligated group Untreated (n = 11), Ischemic group treated with *Eucalyptus* (n = 5), Ischemic group treated with Candesartan (n = 5), and Ischemic group treated with a combination of *Eucalyptus* and Candesartan (n = 10). All experimental rats were kept at the animal house, respecting an alternation of day/night (12 h/12 h). Experimental animals were housed in cages inside a well-ventilated room maintained at 23 ± 1 °C. They were fed a standard diet and water ad libitum, under controlled conditions with regard to the light, temperature, and humidity. All rat groups performed behavioral tests including beam balance, beam walking, modified sticky tape, Morris water maze, and object recognition. Results of the experimental neurological scores were used to compare both the treated and the untreated groups to the control group. The rats were finally sacrificed by cervical dislocation with no pain. All experiments complied with the Animal Testing Ethics of the European Community.

### 4.3. Animal Ligation Operation

Before proceeding with the unilateral ligation of the RCCA, the rat was anesthetized by intraperitoneal injection of ketamine (100 mg/kg). The animal neck was shaved, disinfected, and cleaned with 70% ethyl alcohol. A shallow incision of 1–1.5 cm was made along the rat midline of the mandible base to the sternum. The RCCA was carefully exposed, while avoiding injury to the surrounding soft tissues and nerves. The sternohyoidal and sterno-mastoid muscles were retracted, and the RCCA was ligated with a 4–0 mm silk suture with two knots. Then, the surgical wounds were sutured back with silk. Finally, each rat was returned back to its cage for undergoing behavioral tests once it was fully recovered from anesthesia [43,44]. The operation took 15–20 min for each rat. In our modest laboratory, this surgical procedure is relatively simple. Moreover, the animal mortality rate is low, and the long-term survival is normal.

### 4.4. Animal Treatments

Candesartan Cilexetil (Atacand) was purchased from Takeda Pharmaceutical Company limited, Tokyo, Japan. Candesartan was administered to the rat either alone in a dose of 0.5 mg/kg/day via drinking water or in combination with our laboratory *Eucalyptus* extract. In parallel, our *Eucalyptus* aqueous extract was administered in a dose of 500 mg/kg/day either alone or in combination with Candesartan dissolved in drinking water. The Sham and the Ischemic groups were not both provided with any treatment, only water.

Fourteen days after the surgery, the rats in the sham and untreated groups were provided with water only, with a daily water intake of 100 mL/kg/day [45]. While in the treated group with Candesartan only, the mass of each rat was recorded every morning at 8 a.m. and a dose of 0.5 mg/kg/day was administered by the water according to the mass of each rat.

In parallel, in the *Eucalyptus* group, we introduced the *Eucalyptus* powder (with the dose of 500 mg/kg/day) in the water provided for the rats.

Many studies used different concentrations, but Nishimura et al. showed that the treatment with candesartan at a dose of 0.5 mg/kg/day protected hypertensive rats from brain ischemia by normalizing the cerebral blood flow response [46]. In our present article, we used the latter dose. On the other hand, Yadav et al. and Farhadi et al. both showed that the *Eucalyptus* at a dose of 500 mg/kg/day protects against psychosis in rats and affects growth performance in broiler chicken [47,48].

### 4.5. Body Weight

The body weight of each experimental rat was registered every day due to two main reasons. First, the body weight value is included in the calculation of the amount of *Eucalyptus* and Candesartan to be provided to each rat. Second, it is important to follow-up the health of each experimental rat by observing continuous changes in its body weight.

### 4.6. Plant Collection and Powder Preparation

Leaves of Lebanese *Eucalyptus Camaldulensis* (a member of the family Myrtaceae), with a voucher specimen number 1902, were collected from Hadath, Beirut, Lebanon (at an altitude of 0 m; Coordinates: 33°49′ N 35°31′ E) in February 2018. To eliminate all impurities, the fresh leaves were washed with distilled water and then dried in an oven at 50 °C for two consecutive days. The dried leaves were then manually ground into powder by using mortar and pestle. The powder was finally well-preserved in plastic containers away from humidity, heat, and light.

### 4.7. Preparation of Crude Extracts

Twenty grams of the dried *Eucalyptus* powder was mixed with 100 mL of the selected solvent (water or ethanol). The solution was then put in the sonicator (an ultrasound generating apparatus) at 60 °C for 30 min (min). The extracts were then centrifuged for 5 min at 1200 rpm and poured into a Buchner funnel under reduced pressure to remove insoluble residues. The aqueous extract was initially put at −80 °C, then directly placed in a lyophilizer for 3 days to transform it into powder, while the ethanolic extract was put at −80 °C, subjected first to the Rota-vapor method to remove ethanol, then placed in a lyophilizer. Finally, the dried residue was preserved in plastic falcons in a desiccator and thus ready for use. The obtained powder was then weighed, and the percentage yield was determined according to the following formula:$$Yield\;\left(\%\right)=\frac{\mathrm{Obtained}\;\mathrm{mass}\;\mathrm{of}\;\mathrm{sample}\;\mathrm{extract}}{\mathrm{Initial}\;\mathrm{mass}\;\mathrm{of}\;\mathrm{powder}\;\mathrm{sample}}\times 100$$

### 4.8. Chemical Analyses

In order to study the chemical composition of the prepared plant extracts, a qualitative detection of secondary metabolites was applied (Table 4).

All the chemical reagents used in the phytochemical screening tests were obtained from the pharmaceutical company Merck, Beirut, Lebanon.

### 4.9. Chemical Quantification of Secondary Products

#### 4.9.1. Phenol Content Test

The estimation of the total phenolic content was done using the Folin–Ciocalteau reagent method. Ten mg of the extracted powder was mixed with 10 mL of distilled water for the aqueous extract, and with 10 mL of ethanol for the ethanolic extract. Then, the solution was vortexed for 1 min. Forty μL of the prepared solution was mixed with 3.16 mL of distilled water and 200 μL of Folin reagent. Vortexing was done to obtain solution “A”. Then, 600 μL of 20 % Na_2_CO_3_ was added to solution A, and vortexed for 1 min. The prepared solution was put in a water bath at 40 °C for 30 min. Finally, after it has been removed from the water bath and cooled down, the absorbance of the solution was measured at a wavelength of 765 nm. The results were expressed in mg of Gallic Acid Equivalent (GAE) per g (GAE/g) of dry weight of plant powders.
TPC=GAE×V×D/M
where GAE is the Gallic Acid Equivalence (mg/mL), V is the volume of the extract (mL), D is the dilution factor, and M is the mass (g) of the pure extract of the plant.

#### 4.9.2. Sugar Test

One hundred mg of the *Eucalyptus* powder was macerated in 5.25 mL of 80% ethanol for 12 h, and then centrifuged at 4000 rpm for 10 min. The supernatant containing the sugar was later diluted 50 times. The volume of 0.01 mL of macerated solution is added to 9.99 mL of water to form solution “A”. One g of anthrone (that precipitates the sugar) was dissolved in 500 mL of concentrated sulfuric acid to form solution “B”. 2 mL of solution “A” with 4 mL of solution “B” were mixed and vortexed for 1 min, heated at 92 °C for 8 min, and then placed in the dark for 30 min. Finally, the absorbance is measured by spectrophotometry at 585 nm. The blank tube is prepared similarly, but 2 mL of water was used instead of the solution A.

#### 4.9.3. Dry Matter Test

One g of *Eucalyptus* powder was weighed in a beaker and placed in oven at 100 °C for 24 h. Then, the beaker containing the powder was weighed.
$$\%\mathrm{MF}=\%\mathrm{Dry}\;\mathrm{Matter}=\frac{\left(M2-M0\right)}{\left(M1-M0\right)}\times 100$$
where: M0 = mass of beaker alone, M1 = mass of beaker + powder (before being placed in the oven), and M2 = mass of beaker + powder (after being placed in the oven).

### 4.10. Free Radical 2.2-Diphenyl-1-picrylhydrazyl (DPPH) Scavenging Assay

For each of the two extracts (aqueous and ethanolic), the antioxidant activity (AA) was determined by using the DPPH scavenging assay. 0.0012 g of DPPH was thus dissolved in 50 mL of methanol to obtain a concentration of 6 × 10^−3^ M of DPPH. For the aqueous extract, 5 mg of powder was dissolved in 1 mL of methanol, while for the ethanolic extract, 5 mg of powder was dissolved in a mixture of 0.5 mL of methanol and 0.5 mL of distilled water. Then, 50 μL of the prepared solution was added to 2 mL of DPPH and placed in the dark for 30 min. The absorbance was finally measured at 515 nm. The antioxidant activity was calculated according to the following equation:% Antioxidant activity =100×(ABS control− ABS sample)/ABS control

The absorbance (ABS) control is the absorbance of DPPH + solvent, and ABS sample is the absorbance of the DPPH + sample, where the control was prepared by taking 2 mL of the prepared DPPH solution (DPPH + methanol) without the extract.

### 4.11. Behavioral and Cognitive Tests

#### 4.11.1. Beam Balance Test (BBT)

We performed the BBT in order to assess the sensorimotor integration. We positioned the animal on a wooden beam (0.5 cm wide, 65 cm long, and 45 cm elevated). Taking into consideration the sudden falling of the animal, we placed a foam pad as a cushion underneath the beam. When the rat was securely positioned on the beam, we recorded its performance over 60 s (s) based on the following scale: 1 stands for stable balance, 2 refers to shaky balance, 3 resembles trying to balance by hanging with one or two limbs, 4 reflects trying to balance but falling after 10 s, 5 refers to falling off in under 10 s, and 6 stands for falling off without any effort to balance or hang onto the beam. Three trials should be done for each rat every day. The same procedure was carried out 3 days before the surgery and on days +1, +3, +5, anD + 7 post-surgery [49].

#### 4.11.2. Beam Walking Test (BWT)

BWT is also a sensorimotor test. The rat was trained to walk (on a long narrow and elevated wooden platform (3cm wide, 200cm long, elevated 150cm) from the wooden platform at one extremity (30cm wide, 30cm long) to the darkened goal box at the other extremity (30cm wide, 30cm long). The rat on the platform that was making no attempt to walk was encouraged to react accordingly by prodding (rumping with a pencil or gentle tap on the tail). We assessed the performance of each rat according to the modified scale: score 0, the rat traverses the beam with no foot slip; score 1, the rat grasps on lateral sides of the beam; score 2, the rat shows disability of walking on the beam but can traverse; score 3, the rat takes time to traverse the beam because of walking difficulty; score 4, inability to traverse the beam; score 5, inability to move any limb; score 6, inability to stay on the beam for 10 s [50]. Pre-surgical performance was assessed 3 days before surgery to establish a baseline measure and at post-surgery days +1, +3, +5, anD + 7, with 3 trials for each rat every day. The average daily score and time for each rat were used in the statistical analyses.

#### 4.11.3. Modified Sticky Tape Test (MSTT)

We performed the MSTT in order to assess the sensorimotor function in the rat forelimbs. We used a piece of white paper tape (3 cm long and 1 cm wide) that we wrapped around the carp of the forepaw of the rat so that it could not be removed. A healthy rat would remove it with its mouth vigorously or it might use its contralateral paw. We observed the rat response to the tape for 30 s using two timers: the first one should be placed for a continuous 30 s, and the second one should be turned on only when the rat tried to remove the tape. The test was done for the two limbs separately. We expressed the results as a ratio of values for the left limb to the right. For each rat, three trials were done per day and averaged. The same procedure was done 3 days (D) before the surgery and on days +1, +3, +5, anD + 7 post-surgery [51].

#### 4.11.4. Object Recognition Test (ORT)

The ORT is a memory test that evaluates the rat’s ability to recognize a new object in the environment. The test concept includes the differences in the time of exploration of both objects: novel and familiar [52]. The ORT test consists of habituation, familiarization, and test phases. First, during habituation, we allowed the animal to freely explore the open-field arena for about 5 min, and we returned it to its holding cage. Then, during familiarization, two similar objects (Duplo Toys) must be added to the box at equidistant distance from the rat. The time required to explore each object must be recorded within 5 min. Then, the rat would be put in its cage for 5 min to be returned back to the box, during the test phase, after one of the objects was replaced by a new one having a different shape. Finally, we recorded the time needed to explore the new and old objects within 5 min [53]. This test lasts 20 min for each rat and it was applied at D + 8 post-surgery, and it was very important for qualifying the rats’ short-term memory.

#### 4.11.5. Morris Water Maze Test (MWMT)

The MWMT is used to study long-term spatial learning and memory. It consists of an elevated circular pool (diameter 183 cm and height 38 cm) filled with water and nontoxic yellow powder spices. We divided the pool virtually into 4 quadrants, and we then placed a hidden platform (diameter 13 cm and height 20 cm) at the center of one quadrant. Different cues were fixed around it to guide the rat to navigate and find the platform. The place of the platform changed each day throughout the 7 days of the test (platform in quadrant 1 at D + 10, in quadrant 2 at D + 11 and so on). For each trial, we placed the rat in the pool facing the wall, giving it a maximum of 60 s to reach the submerged platform. A retention time was provided by returning the rat to the holding cage between each placement at a different quadrant position. Before each placement, the rat was given 20 s of exploration on the submerged platform to configure its place with respect to the hanging cues. When the rat failed in finding the platform within 60 s, it was guided to it and put on it for 20 s. Then, we removed the rat and placed it in front of a stove to dry. Finally, we returned the rat to its cage. Each rat underwent 3 trials per day on 3 consecutive quadrants from D + 10 to D + 16 post-surgery [54].

### 4.12. Brain Staining With 2,3,5-Triphenyltetrazolium Chloride (TTC)

After performing all the needed experimental steps according to our study timeline, the rats were sacrificed, and their corresponding brains were removed and directly frozen at −20 °C for 40 min to facilitate cutting. Then, the brains were cut by a sharp blade in sections of 2 mm thickness. Each section was placed in a glass petri dish containing 2% TTC dissolved in normal saline. Petri dishes were covered with aluminum foil to avoid light exposure and placed in an incubator at 37 °C for 30 min. The TTC solution was later replaced by 10% buffered formalin after washing twice with saline solution, and then stained brain sections were photographed [55]. Morphometric analysis was performed using ImageJ software (free download at https://imagej.nih.gov/ij/download.html). Brain damages that stained white in brain sections were counted and their corresponding areas were measured.

### 4.13. Data Analyses

Data was statistically analyzed using the IBM SPSS Statistics 22 Program. Data were subjected to analysis of variance (ANOVA) by the parametric test followed by post hoc Tukey’s test. Differences were considered significant when the *p*-value was less than 0.05.

## 5. Conclusions

The chronic stroke rat model showed its severity on both motor and cognitive functions, however, brain plasticity ensured recovery. The combination therapy that our laboratory team used showed effectiveness to shorten the time to recovery, mainly at the level of the cognitive deficits that are more pronounced than the motor impairment.

## Figures and Tables

**Figure 1 molecules-26-00839-f001:**
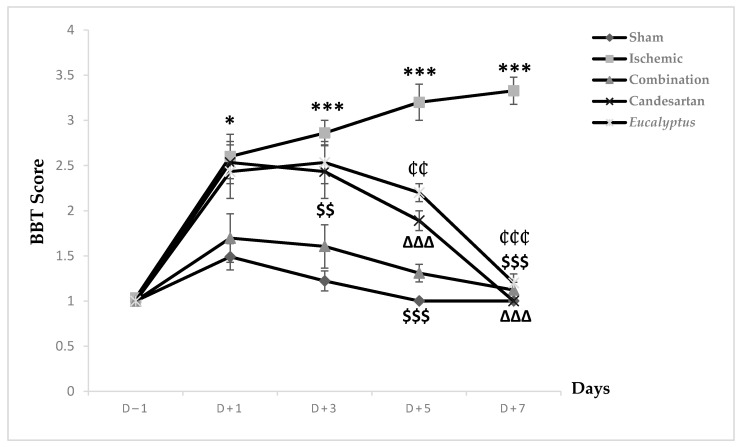
Beam balance test (BBT) scores. BBT scores on days (D) (–1, +1, +3, +5, +7) for the different experimental groups. Sham (n = 6), Ischemic (n = 11), Ischemic treated with a combination of Candesartan and *Eucalyptus* (n = 10), Ischemic treated with Candesartan (n = 5), and Ischemic treated with *Eucalyptus* (n = 5). * *p* < 0.05; *** *p* < 0.001 for a significant difference between Ischemic and Sham; $$ *p* < 0.01; $$$ *p* < 0.01 for a significant difference between Ischemic and Ischemic treated with a combination of Candesartan and *Eucalyptus*; ∆∆∆ *p* < 0.001 for a significant difference between Ischemic and Ischemic treated with Candesartan only; ₵₵ *p* < 0.01 for a significant difference between Ischemic and Ischemic treated with *Eucalyptus* only; ₵₵₵ *p* < 0.01 for a significant difference between Ischemic and Ischemic treated with *Eucalyptus* only.

**Figure 2 molecules-26-00839-f002:**
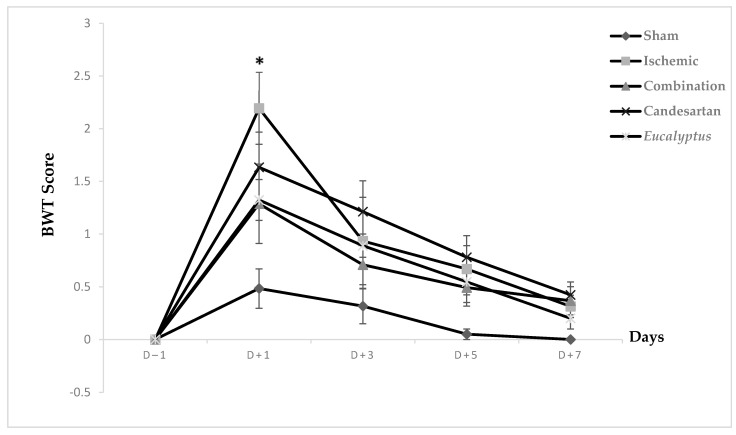
Beam walking test (BWT) scores. BWT results showing the average score on days (−1, +1, +3, +5, +7) for Sham (n = 6), Ischemic (n = 11), Ischemic treated with a combination of Candesartan and *Eucalyptus* (n = 10), Ischemic treated with Candesartan (n = 5), and Ischemic treated with *Eucalyptus* (n = 5) groups. * *p* < 0.05 for a significant difference between Ischemic and Sham groups.

**Figure 3 molecules-26-00839-f003:**
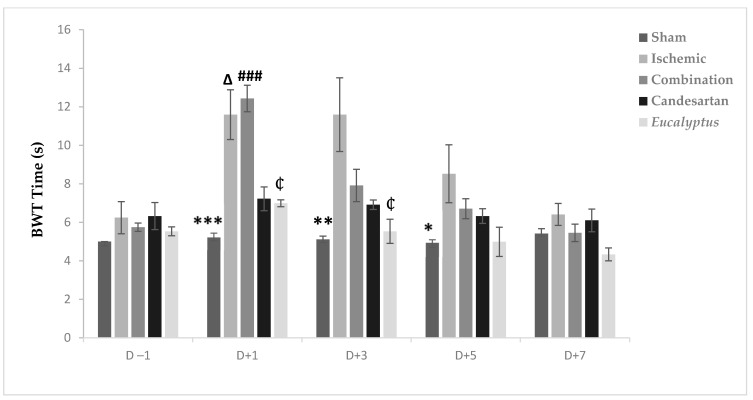
BWT results. BWT results showing the average time to traverse the beam done on days (−1, +1, +3, +5, +7) for Sham (n = 6), Ischemic (n = 11), Ischemic treated with a combination of Candesartan and *Eucalyptus* (n = 10), Ischemic treated with Candesartan (n = 5), and Ischemic treated with *Eucalyptus* (n = 5) groups. **p* < 0.05; ** *p* < 0.01; *** *p* < 0.001 for a significant difference between Ischemic and Sham group; ₵ *p* < 0.05 for a significant difference between Ischemic and Ischemic treated with *Eucalyptus* only; ∆ *p* < 0.05 for a significant difference between Ischemic and Ischemic treated with Candesartan only; ### *p* < 0.001 for a significant difference between Sham and Ischemic treated with combination of Candesartan and *Eucalyptus*.

**Figure 4 molecules-26-00839-f004:**
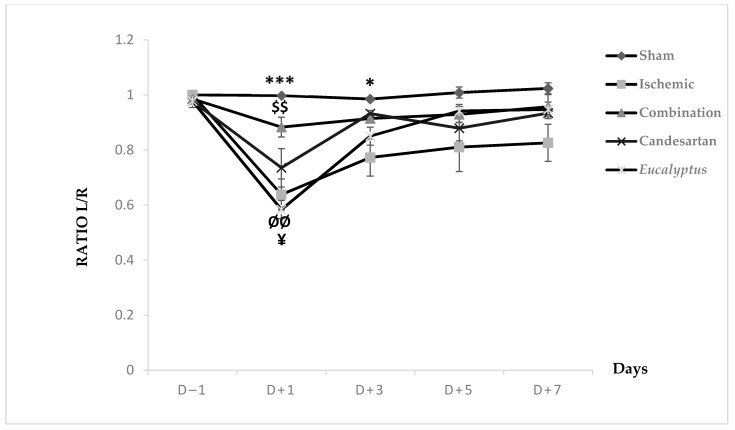
Modified sticky tape test (MSTT) results. MSTT graph showing Left to Right (L/R) ratio of time for scotch sensation done on days (−1, +1, +3, +5, +7) for Sham (n = 6), Ischemic (n = 11), Ischemic treated with a combination of Candesartan and *Eucalyptus* (n = 10), Ischemic treated with Candesartan (n = 5), and Ischemic treated with *Eucalyptus* (n = 5) groups. $$ *p* < 0.01 for a significant difference between Ischemic and Ischemic treated with Combination of Candesartan and *Eucalyptus*; * *p* < 0.05; *** *p* < 0.001 significant difference between Ischemic and Sham; ¥ *p* < 0.05 for a significant difference between Ischemic treated with Combination and *Eucalyptus* only; ØØ *p* < 0.01 for a significant difference between Sham and *Eucalyptus* only.

**Figure 5 molecules-26-00839-f005:**
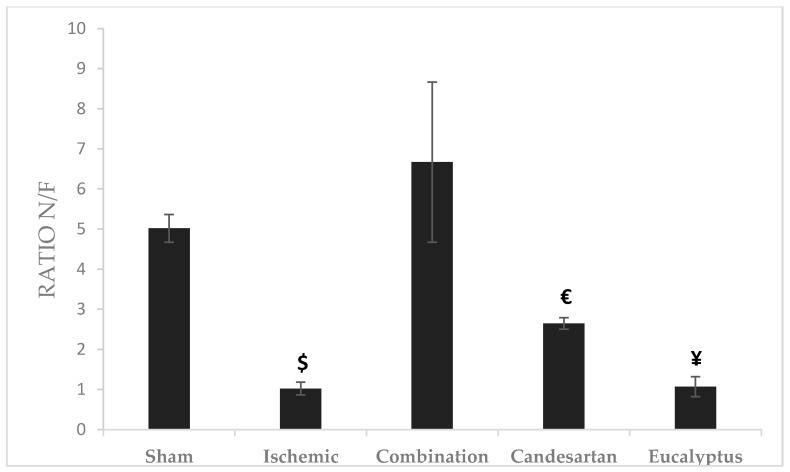
Object recognition test (ORT) results. ORT including novel/familiar (N/F) objects ratio for the time of exploration of a novel object (N) to a familiar one (F) done at D + 8 for Sham (n = 6), Ischemic (n = 11), Ischemic treated with a combination of Candesartan and *Eucalyptus* (n = 10), Ischemic treated with Candesartan (n = 5), and Ischemic treated with *Eucalyptus* (n = 5) groups. $ *p* < 0.05 for a significant difference between Ischemic and Ischemic treated with a combination of Candesartan and *Eucalyptus*; € *p* < 0.05 for a significant difference between Ischemic treated with Combination and Candesartan only; ¥ *p* < 0.05 for a significant difference between Ischemic treated with Combination and *Eucalyptus* only.

**Figure 6 molecules-26-00839-f006:**
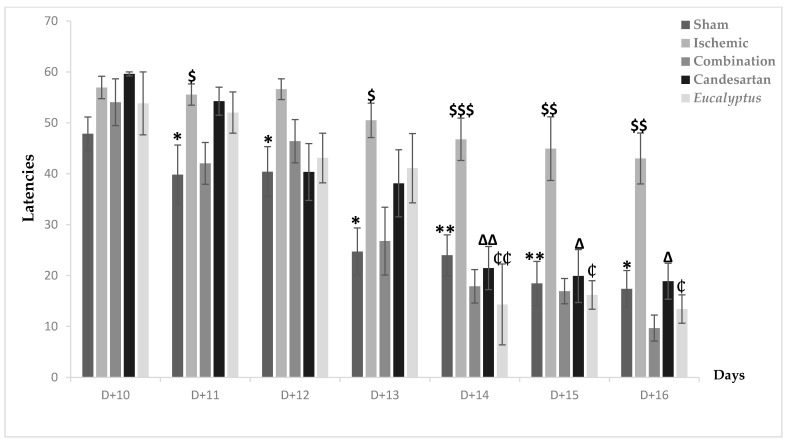
Morris water maze test (MWMT) results. MWMT results showing average escape latencies starting from D + 10 until D + 16 for Sham (n = 6), Ischemic (n = 11), Ischemic treated with a combination of Candesartan and *Eucalyptus* (n = 10), Ischemic treated with Candesartan (n = 5), and Ischemic treated with *Eucalyptus* (n = 5) groups. * *p* < 0.05; ** *p* < 0.01 for a significant difference between Ischemic and Sham groups; $ *p* < 0.05; $$ *p* < 0.01; $$$ *p* < 0.001 for a significant difference between Ischemic and Ischemic treated with a combination of Candesartan and *Eucalyptus*; ∆ *p* < 0.05; ∆∆ *p* < 0.01 for a significant difference between Ischemic and Ischemic treated with Candesartan only; ₵ *p* < 0.05; ₵₵ *p* < 0.01 for a significant difference between Ischemic and Ischemic treated with *Eucalyptus* only.

**Figure 7 molecules-26-00839-f007:**
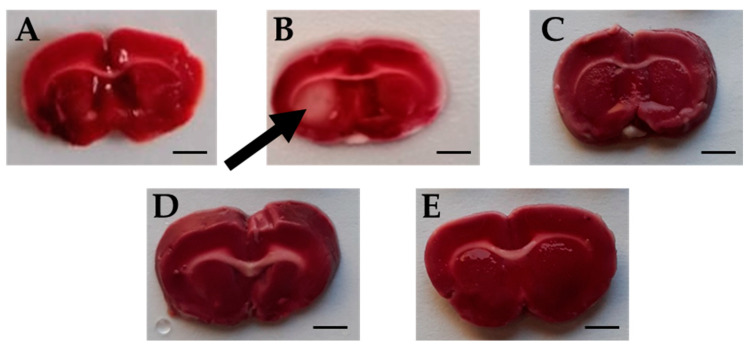
Rat brain sections stained by triphenyltetrazolium chloride (TTC). Images of brain sections (**A**–**E**) correspond to Sham (n = 6), Ischemic (n = 11), Combination (n = 10), Candesartan (n = 5), and *Eucalyptus* (n = 5) groups respectively, 14 days after right common carotid artery ligation. Images shown are representative of sections observed for each group. White brain area corresponding to damages caused by ischemia is indicated by an arrow. The red staining sections corresponded to normal non-ischemic areas. Bar scale = 3 mm.

**Figure 8 molecules-26-00839-f008:**
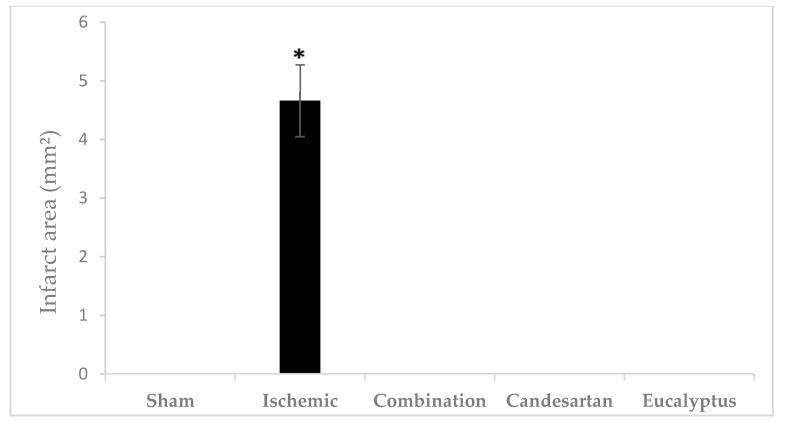
Cerebral infarct area. Cerebral infarct area of the rat brain sections stained by TTC 14 days after right common carotid artery ligation for Sham (n = 6), Ischemic (n = 11), Ischemic treated with a combination of Candesartan and *Eucalyptus* (n = 10), Ischemic treated with Candesartan (n = 5), and Ischemic treated with *Eucalyptus* (n = 5) groups. * *p* < 0.05 for a significant difference between Ischemic and Sham groups. The bar represents the mean ± SEM (Standard error of the mean).

**Table 1 molecules-26-00839-t001:** Extraction yields of *Eucalyptus Camaldulensis* crude extract with two different solvents.

Solvent	Initial Weight (g)	Final Weight (g)	Extraction Yield (%)
Ethanol	20.0	3.3	16.5
Water	20.0	4.0	20.0

**Table 2 molecules-26-00839-t002:** Phytochemical screening tests of the ethanolic and aqueous extracts of *Eucalyptus Camaldulensis.*

Plant Constituents	Ethanol	Aqueous
Reducing sugar	+	+
Anthraquinones	+	+
Proteins and amino acids	−	−
Phlabotannins	−	−
Alkaloids	+	+
Tannins	+	+
Resins	+	−
Terpenoids	+	+
Flavonoids	+	+
Quinones	+	+
Sterols and Steroids	+	+
Diterpenes	+	+
Anthocyanins	−	−
Flavanones	+	+
Lignines	+	+
Cardiac glycosides	+	−
Saponins	−	+
Phenols	+	+
Fixed oils	+	+

**Table 3 molecules-26-00839-t003:** Chemical traits of two different extracts of *Eucalyptus*
*Camaldulensis.*

Solvent	TPC (mg Gallic Acid Equivalence (GAE)/g of Extract)	Soluble Sugar Test (%)	Dry Matter (%)	Antioxidant Activity (Trolox Equivalents (TE) Umol/g of Extract)
Ethanol	15.33 ± 0.54	9.95	95.06	211.89 ± 0.39
Aqueous	19.52 ± 0.22	9.95	95.06	214.01 ± 1.66

**Table 4 molecules-26-00839-t004:** Phytochemical screening tests.

Plant Constituents	Extract	Reagent	Color
Reducing sugar	0.5 mL	1 mL water + 5–8 drops of Fehlings	Brick-red precipitate
Anthraquinones	1 mL	1 mL HCL (10%) + boil	Precipitate
Proteins and amino acids	1 mL	1 mL ninhydrin (0.25%) + boil	Blue color
Phlabotannins	1 mL	1 mL HCL (1%) + boil (5 min) + cooling	Red precipitate
Alkaloids	1 mL	5 drops of Dragendorff	Reddish orange precipitate/reddish brown or turbid
Tannins	1 mL	Ferric chloride FeCl_3_ (1%)	Blue color
Resins	1 mL	Acetone + small amount of water + agitation	Turbidity
Terpenoids	1 mL	2 mL chloroform + 3 mL concentrated sulfuric acid	Reddish brown color in the surface
Flavonoids	1 mL	5 mL potassium hydroxide KOH (50%)	Yellow color
Quinones	1 mL	HCL concentrated	Precipitate or yellow color
Sterols and Steroids	1 mL	2 mL chloroform + concentrated sulfuric acid	Red color of the upper layer + greenish yellow fluorescence in the acid layer
Diterpenes	1 mL dissolved in water	Few drops of copper sulphate	Green color
Anthocyanins	1 mL	1 mL NaOH (10%)	Blue color
Flavanones	1 mL	1 mL concentrated sulfuric acid	Purple red color
Lignines	2 mL	Safranine	Pink color
Cardiac glycosides	2 mL	1 mL acetic acid glacial + 1 drop ferric chloride FeCl_3_ (5%) + 1 mL concentrated sulfuric acid	Purple ring + brown ring + green ring
Saponins	2 mL	Vigorous shaking (5 min on Vortex)	Layer of foam
Phenols	5 mL	1 mL FeCl_3_ (1%) + 1 mL K_3_(Fe(CN)_6_) (1%)	Greenish blue color
Fixed oils and Fatty acids	Small amount of extract	On filter paper	Oil spot

## Data Availability

The study did not report any data.

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
