# Peer review of "Study of Neuroprotection by a Combination of the Biological Antioxidant (Eucalyptus Extract) and the Antihypertensive Drug Candesartan against Chronic Cerebral Ischemia in Rats"

_molecules, 2021, doi:10.3390/molecules26040839_

Round 1
Reviewer 1 Report
The manuscript titled as ‘Study Of Neuroprotection By A Combination Of The Biological Antioxidant (Eucalyptus Extract) And The Antihypertensive Drug Candesartan Against Chronic Cerebral Ischemia In Rats’ by Trabolsi et al. verified, shows that the combination of Eucalyptus and Candesartan could decrease ischemic brain injury and improve neurological outcomes. This work should be of wide interests to most researchers on neuroscience and molecular medicine etc.
This manuscript has sufficient novel and findings and the method described is highly practical. I recommend that the manuscript be accepted with some revisions. The following points need to be addressed:
- The authors need to explain why they administered Candesartan at a dose of 0.5 mg/kg/day, whereas Eucalyptus aqueous extract at a dose of 500 mg/kg/day. Any supporting references, or an experimental test of the dose-dependent effect may be needed/
- For the part of '3.5.7. Triphenyltetrazolium Chloride (TTC) Test' and Figure 9, is it possible to quantiby the TTC staining, and the size rat brain sections under different conditions?
Author Response
Dear Reviewer, Dear Editor,
we thank you for your comments and the chance to submit a revised version of our work.
Please find our answers to Reviewer 1:
The manuscript titled as ‘Study Of Neuroprotection By A Combination Of The Biological Antioxidant (Eucalyptus Extract) And The Antihypertensive Drug Candesartan Against Chronic Cerebral Ischemia In Rats’ by Trabolsi et al. verified, shows that the combination of Eucalyptus and Candesartan could decrease ischemic brain injury and improve neurological outcomes. This work should be of wide interests to most researchers on neuroscience and molecular medicine etc.
We thank you for your invaluable and rewarding comment.
This manuscript has sufficient novel and findings and the method described is highly practical. I recommend that the manuscript be accepted with some revisions. The following points need to be addressed:
- The authors need to explain why they administered Candesartan at a dose of 0.5 mg/kg/day whereas Eucalyptus aqueous extract at a dose of 500 mg/kg/day. ( Any supporting references, or an experimental test of the dose-dependent effect may be needed/
We thank you for your comments that they had undoubtedly a good impact on our research work.
(Page 3; L.116-118) Many studies used different concentrations but Nishimura et al. showed that the treatment with candesartan at a dose of 0.5 mg/kg/day protected hypertensive rats from brain ischemia by normalizing the cerebral blood flow response.
- Nishimura, Y.; Ito, T.; Saavedra, J.M. Angiotensin II AT(1) blockade normalizes cerebrovascular autoregulation and reduces cerebral ischemia in spontaneously hypertensive rats. Stroke2000, 31, 2478–2486). In our present article, we used the latter dose.
(Page 3; L.119-121) On the other hand, Yadav et al. and Farhadi et al. both showed that the Eucalyptus at a dose of 500 mg/kg/day protects against psychosis in rats and affects growth performance in broiler chicken.
- Yadav, M.;Jindal, D.K.;Parle, M.;Kumar, A.; Dhingra, S. Targeting oxidative stress, acetylcholinesterase, proinflammatory cytokine, dopamine and GABA by eucalyptus oil (Eucalyptus globulus) to alleviate ketamine-induced psychosis in rats.Inflammopharmacology 2019, 27, 301-311. DOI: 10.1007/s10787-018-0455-3.
- Farhadi, D.;Karimi, A.;Sadeghi, G.;Sheikhahmadi, A.;Habibian, M.;Raei, A.; Sobhani, K. Effects of using eucalyptus (Eucalyptus globulus) leaf powder and its essential oil on growth performance and immune response of broiler chickens.Iran J Vet Res 2017, 18, 60-62.
- For the part of '3.5.7. Triphenyltetrazolium Chloride (TTC) Test' and Figure 9, is it possible to quantify the TTC staining, and the size rat brain sections under different conditions?
We thank you for your comment and we took it into consideration in our revised article.
Regards
Reviewer 2 Report
I have read the manuscript entitled “Study of neuroprotection by a combination of the biological antioxidant (Eucalyptus extract) and the antihypertensive drug candesartan against chronic cerebral ischemia in rats” and below I give detailed comments and suggestions.
- English definitely needs professional correction.
- Materials and methods:
a). The Authors performed the unilateral ligation of the right common carotid artery as a model of chronic cerebral ischemia. Can the Authors explain what this is the kind of model of stroke? Who previously did such model of animal stroke perform (give the citations, please)?
b). It is well established that excitotoxicity, a type of neurotoxicity evoked by elevated extracellular glutamate level is a primary contributor to ischemic neuronal death (Choi, 1994; Lai et al., 2014). Since it is known that ketamine, a potent non-competitive NMDA receptor antagonist decreased the glutamate-induced excitotoxicity and produced neuroprotection against cerebral ischemic injury in rodents (Xiong et al., 2020), in light of the above can the Authors explain why they anesthetized rats witch ketamine (100 mg/kg)? What was time from animal operation to doing experiments?
c). The Authors should describe schedule of the drugs treatment and in this paragraph present also information on the basis of whose studies the doses of the drugs were chosen.
- Results:
a). The Authors should improve Fig. 9 because the representative TTC staining of coronal sections of the rat brain are unclear with poor resolution. Moreover where are the results of a morphometric analysis of the infarct volume after cerebral ischemia or ischemia + drugs treatment, calculated from TTC coronal sections of the brain.
- The Discussion have been written very chaotically. There are a lot of lack of citations of other authors (for example page 15, line 490 “previous study”.. Whose previous studies??
Page 16, lines 512-514 The Authors wrote that “According to previous studies, Candesartan treatment has been shown to improve sensorimotor and cognitive function after common carotid artery ligation in rats and tested by rotarod and MWMT” and cited paper by Villapol et al., (2012) entitled “Candesartan, an angiotensin II AT₁-receptor blocker and PPAR-γ agonist, reduces lesion volume and improves motor and memory function after traumatic brain injury in mice”. The Authors should cite the proper literature.
Page 17, lines 564-566 The Authors wrote that “Results indicated that the histological damage measured by TTC reached their maximum 24h after stroke in a model of carotid artery ligation in the mouse”. This sentence is unclear. Can the Authors explain in which context this sentence is?
Author Response
Dear Reviewer, Dear Editor,
We would like to thank you for your constructive comments.
Please find our answers below:
Reviewer 2 :
I have read the manuscript entitled “Study of neuroprotection by a combination of the biological antioxidant (Eucalyptus extract) and the antihypertensive drug candesartan against chronic cerebral ischemia in rats” and below I give detailed comments and suggestions.
We sincerely thank you for your important pieces of advice and constructive comments.
- English definitely needs professional correction.
We have corrected the grammatical mistakes. An English revision has been made and changes had been conducted.
- Materials and methods:
a). The Authors performed the unilateral ligation of the right common carotid artery as a model of chronic cerebral ischemia. Can the Authors explain what this is the kind of model of stroke? Who previously did such model of animal stroke perform (give the citations, please)?
We thank you for your important comment.
(Page 3; L. 100-102) In our modest laboratory, this surgical procedure is relatively simple. Moreover, the animal mortality rate is low and the long-term survival is normal.
Cao et al. and as well as Chen et al. used previously such a model.
- Cao, D.;Bai, Y.; Li, L. Common Carotid Arteries Occlusion Surgery in Adult Rats as a Model of Chronic Cerebral Hypoperfusion.Bio-protocol 2018, 8, e2704. DOI: 10.21769/BioProtoc.2704.
- Chen, S.T.;Hsu, C.Y.;Hogan, E.L.;Maricq, H.; Balentine, J.D. A model of focal ischemic stroke in the rat: reproducible extensive cortical infarction.Stroke 1986, 17, 738-43. DOI: 10.1161/01.str.17.4.738.
b). It is well established that excitotoxicity, a type of neurotoxicity evoked by elevated extracellular glutamate level is a primary contributor to ischemic neuronal death (Choi, 1994; Lai et al., 2014). Since it is known that ketamine, a potent non-competitive NMDA receptor antagonist decreased the glutamate-induced excitotoxicity and produced neuroprotection against cerebral ischemic injury in rodents (Xiong et al., 2020), in light of the above can the Authors explain why they anesthetized rats witch ketamine (100 mg/kg)? What was time from animal operation to doing experiments?
We thank you for this very important comment.
The neuroprotective effect of ketamine is not a variable factor in our work because the ketamine was injected to all groups. The ketamine did not show a protection effect in the rats in the untreated group. The operation took 15-20 minutes for each rat (Page 3; L.100).
c). The Authors should describe schedule of the drugs treatment and in this paragraph present also information on the basis of whose studies the doses of the drugs were chosen.
Thank you for your important comment.
(Page 3; L.110-121) Fourteen days after the surgery, the rats in the sham and untreated groups were provided with water only. While in the treated group with Candesartan only, the mass of each rat was recorded every morning at 8 a.m. and a dose of 0.5 mg/kg/day was administered by the water according to the mass of each rat.
In parallel, in the Eucalyptus group, we introduced the Eucalyptus powder (with the dose of 500 mg/kg/day) in the water provided for the rats.
- Results:
a). The Authors should improve Fig. 9 because the representative TTC staining of coronal sections of the rat brain are unclear with poor resolution. Moreover where are the results of a morphometric analysis of the infarct volume after cerebral ischemia or ischemia + drugs treatment, calculated from TTC coronal sections of the brain.
We thank you for your comment. We changed the image and we did the morphometric analysis.
- The Discussion have been written very chaotically. There are a lot of lack of citations of other authors (for example page 15, line 490 “previous study”. Whose previous studies??
Page 16, lines 512-514 The Authors wrote that “According to previous studies, Candesartan treatment has been shown to improve sensorimotor and cognitive function after common carotid artery ligation in rats and tested by rotarod and MWMT” and cited paper by Villapol et al., (2012) entitled “Candesartan, an angiotensin II AT₁-receptor blocker and PPAR-γ agonist, reduces lesion volume and improves motor and memory function after traumatic brain injury in mice”. The Authors should cite the proper literature.
We thank you for your comment.
The section has been changed accordingly and the citations were added.
Page 17, lines 564-566 The Authors wrote that “Results indicated that the histological damage measured by TTC reached their maximum 24h after stroke in a model of carotid artery ligation in the mouse”. This sentence is unclear. Can the Authors explain in which context this sentence is?
We thank you for your comment.
The above sentence describes a research study that showed that the results of TTC can appear 24h after the carotid artery occlusion, we added accordingly the related reference and improved the English.
regards
Round 2
Reviewer 2 Report
I have read the revision version of the manuscript entitled “Study of neuroprotection by a combination of the biological antioxidant (Eucalyptus extract) and the antihypertensive drug candesartan against chronic cerebral ischemia in rats” and below I give my comments.
- The Authors still have not presented the results of a morphometric analysis of the infarct volume after cerebral ischemia or ischemia + drugs treatment, calculated from TTC coronal sections of the brain. Where is the graph presenting the effect of ischemia and ischemia treated with a Combination of Candesartan and Eucalyptus with statistical significant differences?
- Where is description of the Figure 8?
- The Authors still have not improved the Discussion. Moreover, in my last review I noted that the Authors should cite the proper literature. Why did the Authors write that “According to previous studies, Candesartan treatment has been shown to improve sensorimotor and cognitive function after common carotid artery ligation in mice and tested by rotarod and MWMT” and have cited paper describing model of traumatic brain injury, but not model of carotid artery ligation?
Author Response
Dear Reviewers,
Thank you for reviewing our manuscript entitled « Study Of Neuroprotection By A Combination Of The Biological Antioxidant (Eucalyptus Extract) And The Antihypertensive Drug Candesartan Against Chronic Cerebral Ischemia In Rats ». We appreciate that you have provided us with the opportunity to resubmit our revised paper based on the comments of the referees.
Please find below our response to your comments. We feel that we have addressed all the reviewers’ comments and updated the manuscript accordingly by including additional analyses. Kindly find below a point-by-point response to reviewers’ comments.
Yours sincerely,
Christine Trabolsi,
On behalf of all authors.
Reviewers’ comments:
Point 1 : « The Authors still have not presented the results of a morphometric analysis of the infarct volume after cerebral ischemia or ischemia + drugs treatment, calculated from TTC coronal sections of the brain. Where is the graph presenting the effect of ischemia and ischemia treated with a Combination of Candesartan and Eucalyptus with statistical significant differences? »
Answer 1: The graph asked by the Reviewer has been added (new Figure 9). The statistical significancy has been calculated and shown in Fig 9.
Point 2 : « Where is description of the Figure 8? »
Answer 2: The description has been added.
Point 3: “The Authors still have not improved the Discussion. Moreover, in my last review I noted that the Authors should cite the proper literature. Why did the Authors write that “According to previous studies, Candesartan treatment has been shown to improve sensorimotor and cognitive function after common carotid artery ligation in mice and tested by rotarod and MWMT” and have cited paper describing model of traumatic brain injury, but not model of carotid artery ligation?”
Answer 3 : We thank you for your comment.
The section has been changed accordingly and the missing citation was added (L.535 and 537). According to previous studies, Candesartan treatment has been shown to improve sensorimotor and cognitive function after traumatic brain injury (TBI) in mice and tested by rotarod and MWMT [41]. Studies show that Aliskiren, Enalapril and Candesartan significantly improve spatial learning and memory and inhibit hippocampal apoptosis [42].
Regards